# Chromatin-mediated translational control is essential for neural cell fate specification

Dong-Woo Hwang[1,2], Anbalagan Jaganathan[1], Padmina Shrestha[1,3], Ying Jin[1], Nour El-Amine[1], Sidney H Wang[4], Molly Hammell[1], Alea A Mills[1]

**Neural cell fate specification is a multistep process in which stem cells undergo sequential changes in states, giving rise to particular lineages such as neurons and astrocytes. This process is accompanied by dynamic changes of chromatin and in transcription, thereby orchestrating lineage-specific gene expression programs. A pressing question is how these events are interconnected to sculpt cell fate. We show that altered chromatin due to loss of the chromatin remodeler Chd5 causes neural stem cell activation to occur ahead of time. This premature activation is accompanied by transcriptional derepression of ribosomal subunits, enhanced ribosome biogenesis, and increased translation. These untimely events deregulate cell fate decisions, culminating in the generation of excessive numbers of astrocytes at the expense of neurons. By monitoring the proneural factor Mash1, we further show that translational control is crucial for appropriate execution of cell fate specification, thereby providing new insight into the interplay between transcription and translation at the initial stages of neurogenesis.**

## Introduction

In both the embryonic and the adult brain, specialized astrocytes residing in germinal niches of the ventricular–subventricular zone surrounding the lateral ventricles and the subgranular zone of the dentate gyrus of the hippocampus are recognized as brain-specific neural stem/progenitor cells (NSCs) (Adams et al, 2004; Kriegstein & Alvarez-Buylla, 2009; Bayraktar et al, 2014; Taverna et al, 2014; Bond et al, 2015). These neuroepithelial cell derivatives with features of the astroglial lineage are also referred to as radial glial cells and type B cells in the embryonic and adult brain, respectively, based on their apicobasal cellular architecture and expression of astroglial markers (Alvarez-Buylla et al, 2001; Merkle et al, 2004). These NSCs persist throughout adult life, retaining the capacity to give rise to the major lineages of the brain, including neurons and glia (Gage & Temple, 2013).

Specification of NSC fate is fundamentally tied to the question of how a cell exits from the uncommitted state of the multipotent stem cell, transitions to a more restricted state of the immediate progenitor cell, and ultimately becomes a terminally differentiated cell with a specified fate—that is, what intrinsic and/or extrinsic molecular mechanisms underlie this process? (Morrison et al, 1997; Edlund & Jessell, 1999). Inherent heterogeneity of the neural progenitor pool—which is a feature of both the developing and the postnatal mouse brain—has been proposed as a source of diversity for neuronal and glial cell types (Alvarez-Buylla et al, 2008; Rowitch & Kriegstein, 2010; Ihrie & Alvarez-Buylla, 2011). In particular, the observation that different types of adult olfactory bulb interneurons originate from distinct progenitor populations exemplifies how progenitor heterogeneity can provide an intrinsic mechanism that drives neural cell fate specification (Merkle et al, 2007; Alvarez-Buylla et al, 2008). Recent findings for coexistence of a population of slowly cycling quiescent NSCs (i.e., "quiescent" NSCs, qNSCs) with a population of a mitotically active NSCs (i.e., "activated" NSCs, aNSCs) in the ventricular–subventricular and the subgranular zones of adult mice further highlight inherent heterogeneity at the transcriptional level, providing a complex picture of how transcriptional regulation underlies neural cell fate specification (Codega et al, 2014; Mich et al, 2014; Llorens-Bobadilla et al, 2015; Shin et al, 2015). Despite this evidence supporting the interconnection between stem cell fate and transcription, the direct or indirect roles of chromatin remodelers and the extent to which the changes exerted by these remodelers affect chromatin dynamics, transcriptional cascades, and downstream events driving cell fate specification have yet to be clearly understood.

The chromatin remodeling protein CHD5 (Li et al, 2014; Quan & Yusufzai, 2014) is a tumor suppressor encoded at human 1p36 (Bagchi et al, 2007)—a genomic region frequently deleted in a variety of cancers, including neuroblastoma and glioma (Bagchi & Mills, 2008). Chd5 is expressed robustly in terminally differentiated postmitotic neurons (Vestin & Mills, 2013) and is also detected in

[1]Cold Spring Harbor Laboratory, Cold Spring Harbor, NY, USA   [2]Graduate Program in Genetics, Stony Brook University, Stony Brook, NY, USA   [3]Molecular and Cellular Biology Program, Stony Brook University, Stony Brook, NY, USA   [4]Center for Human Genetics, The Brown Foundation Institute of Molecular Medicine, The University of Texas Health Science Center at Houston, Houston, TX, USA

Correspondence: mills@cshl.edu
Dong-Woo Hwang's present address is Howard Hughes Medical Institute Janelia Research Campus, Ashburn, VA, USA.

hippocampal progenitor cells expressing Sox3 (Egan et al, 2013)—a transcription factor that marks proliferating and differentiating neural progenitors as well as some populations of postmitotic neurons of the forebrain (Wang et al, 2006). Chd5 is also expressed in neural progenitors isolated from embryonic day 14.5 (E14.5) rat cortices (Nitarska et al, 2016), suggesting that it plays a role in early stages of neurogenesis. It has been reported that Chd5 knockdown during late neurogenesis impedes cortical migration (Egan et al, 2013; Nitarska et al, 2016). However, it is not clear whether Chd5 functions at early stages to restrict the NSC transition to the mitotically active state, and if so, how perturbation of this bottleneck affects cell fate specification at later stages of neurogenesis. Using primary neural stem/progenitor cells isolated from Chd5-deficient neonatal brains, we found that untimely NSC activation at the beginning of differentiation alters cell fate decisions to favor the astroglial over the neuronal lineage. These findings indicate that the NSC activation process at the very onset of differentiation is crucial for fine-tuning ultimate cell fate and that the chromatin remodeler Chd5 plays a pivotal role in this process. We show that prematurely activated Chd5-deficient NSCs have chromatin perturbations and that more than half of the genes encoding ribosomal subunit proteins are significantly up-regulated, indicating that Chd5-mediated chromatin regulation is essential for transcriptional regulation of ribosome biogenesis. Importantly, an increase in protein synthesis driven by this untimely enhancement of ribosome biogenesis disrupts the dynamic translational regulation aberrant expression of the key proneural transcription factor Mash1 at the beginning of differentiation, thereby altering neuronal lineage specification. These observations reveal multiple layers of regulatory control that dictate cell fate specification and demonstrate that the temporal coordination of these events plays a central role in the execution of neural cell fate.

# Results

### Chd5 deficiency causes premature activation of neural stem/progenitor cells

To determine whether Chd5 deficiency impacts inherent stem cell characteristics, we used a previously generated Chd5-deficient mouse model (Li et al, 2014) and examined the morphology and proliferation of primary NSCs isolated from postnatal day 1 (P1) +/+ and *Chd5−/−* brains using previously established protocols (Reynolds & Weiss, 1992; Deleyrolle & Reynolds, 2009). Compared with controls, Chd5-deficient NSCs grown under adherent conditions had a markedly distinct morphology, displaying a stocky shape with less pronounced radial processes (Fig 1A). Chd5-deficient NSCs also had enhanced proliferative potential, as indicated by their ability to form larger neurospheres that incorporated significantly more EdU than controls (Fig 1A–C). Importantly, exogenous Chd5 rescued the enhanced proliferative potential of Chd5-deficient NSCs, effectively reducing the size of the neurospheres (see Fig 1B). This shows that deficiency of Chd5 enhances NSC proliferation and that exogenous Chd5 reverses this effect, thereby indicating that Chd5 is necessary and sufficient for proper regulation of NSC proliferation.

We next asked whether these features of Chd5-deficient NSCs represented a shift from the relatively quiescent to the more activated state by assessing expression of activated NSC (aNSC)-specific markers in early passage +/+ and *Chd5−/−* NSCs. In support of this idea, we found that the aNSC Egfr+ (i.e., Egfr^high) population was significantly expanded in Chd5-deficient NSCs (Fig 1D), whereas the general NSC (i.e., Cd133+) population was not appreciably altered (Fig 1E). Consistent with expansion of the Egfr+ population, the NSC marker paired box 6 (Pax6) was decreased, whereas the aNSC marker nestin—an intermediate filament protein predominantly expressed in aNSCs (Codega et al, 2014; Mich et al, 2014)—was enhanced (Fig 1F and G). Immunofluorescent analysis for nestin expression relative to the general NSC marker vimentin further showed enrichment of the activated NSC population (Fig 1H and I). These findings indicate that Chd5 deficiency promotes precocious mitotic activation and enhanced proliferative potential.

### Premature activation at the onset of differentiation dictates ultimate cell fate

We asked if the premature activation we observed in Chd5-deficient NSCs at the initial stages of the differentiation process would impact cell fate specification at later stages. To test this idea, we first assessed the temporal dynamics of the NSC activation process over the course of differentiation by monitoring nestin expression (Fig S1A). In wild-type NSCs, nestin levels increased immediately upon differentiation, plateaued between 3 and 12 h, and gradually decreased afterwards to levels similar to that of undifferentiated cells by the 24-h time point (Fig S1B). In contrast with the highly dynamic nature of NSC activation we found in control cells, nestin expression increased in far smaller increments in Chd5-deficient NSCs, peaked at 6 h, and maintained this level over the first 24-h period of differentiation (Fig S1C). The degree of change between successive time points was much smaller in Chd5-deficient NSCs, revealing a relatively flat profile when compared with controls (Fig S1D). These findings indicate that both the timing and the extent of stem cell activation at the initial stages of differentiation are deregulated in Chd5-deficient NSCs.

To test whether the premature activation of *Chd5−/−* NSCs affected cell fate specification at later stages, we performed neural differentiation assays and assessed cellular identities of mixed neural lineages (i.e., neurons and astrocytes) at 2, 4, and 7 d post-differentiation (Figs S1E and 2A, B, and E). Strikingly, immunofluorescent microscopy 2 d into the differentiation process revealed that wild-type NSCs generated mostly neurons (i.e., cells expressing the neuron-specific microtubule-associated protein 2 [Map2]) with a smaller fraction of astrocytes (i.e., cells expressing the astrocyte-specific glial fibrillar acidic protein [Gfap]) (Fig S1E). Although Chd5-deficient NSCs also gave rise to both neurons and astrocytes, the distributions of these two lineages were dramatically altered relative to controls, with neurons being notably underrepresented. Consistent with this observation, Map2 levels in P1 neonatal *Chd5−/−* cortex were considerably lower compared with wild-type littermate controls, further confirming underrepresentation of the neuronal lineage (Fig S1F). In addition, we noted that the early neuronal marker, T-box brain gene (Tbr1), was compromised in Chd5-deficient cells at 3-h post-differentiation, whereas the intermediate

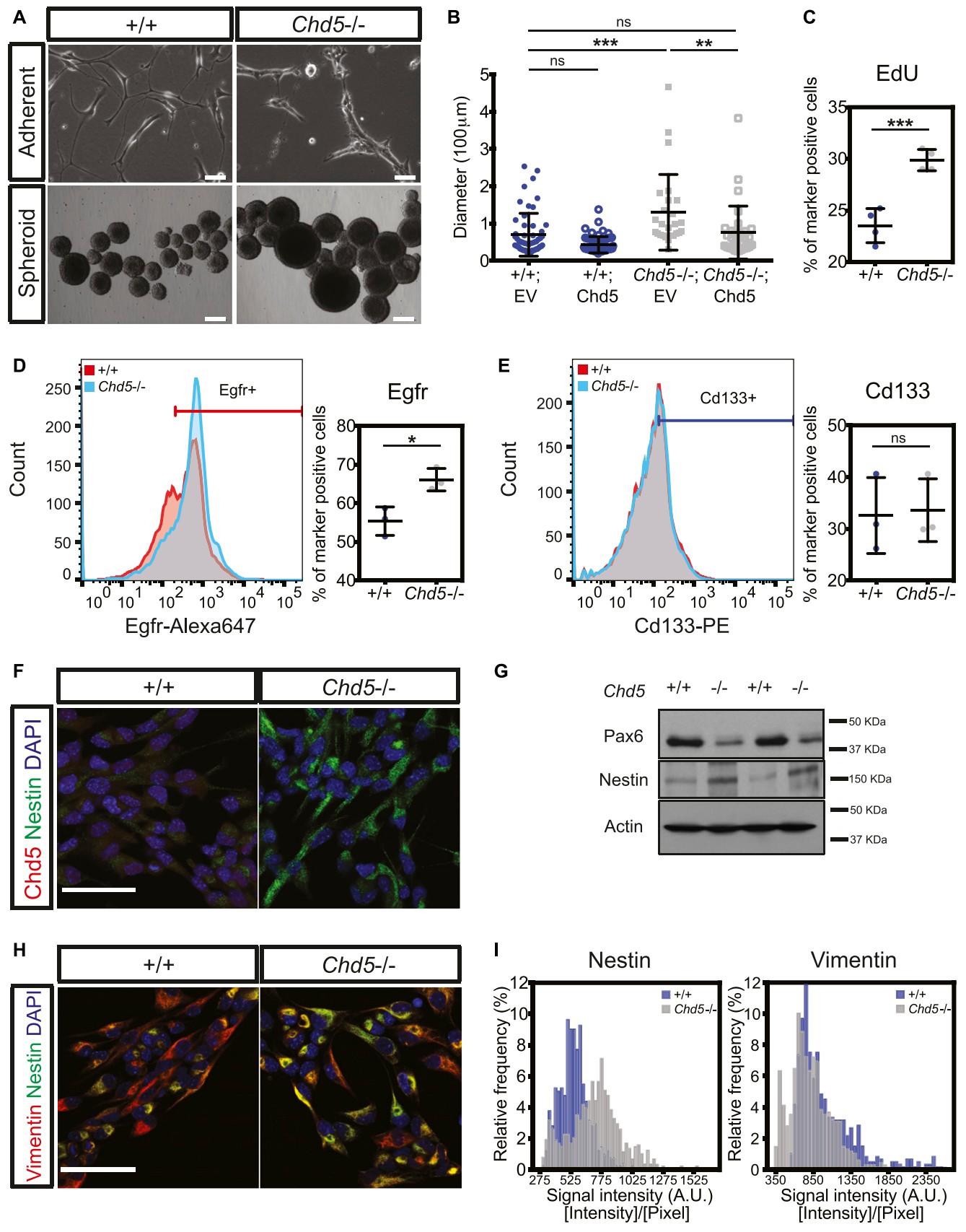

progenitor marker, eomesodermin homolog (Tbr2), did not appreciably change (Fig S1G), suggesting that neuronal lineage specification of Chd5−/− NSCs was affected at an early stage of differentiation. Most importantly, differentiation assays at later time points (i.e., 4 and 7 d post-differentiation) revealed that Chd5-deficient cultures had an underrepresentation of Map2+ neurons, and an overabundance of Gfap+ astrocytes (Fig 2). These findings indicate that Chd5 deficiency and subsequent premature activation of NSCs at the onset of differentiation causes misdirected cell fate specification at later stages.

To determine whether altered cell fate specification in Chd5-deficient NSCs was caused by Chd5's absence, we reintroduced exogenous Chd5 and assessed cell fate specification 7 d into the differentiation process (see Fig 2C, D, and F). This revealed that Chd5-deficient NSCs expressing exogenous Chd5 generated similar ratios of neurons to astrocytes as did controls, confirming a causal link between Chd5 and cell fate specification. These findings demonstrate that Chd5-deficient NSCs undergo altered cell fate decisions, favoring the astroglial over the neuronal lineage, and that reintroduction of exogenous Chd5 is sufficient to correct these cell fate specification defects. Thus, the timing and degree of NSC activation at the very onset of differentiation fine-tunes ultimate cell fate, and the chromatin remodeler Chd5 plays a pivotal role in this process.

### Chd5 deficiency affects chromatin and causes large-scale transcriptional derepression of genes encoding ribosomal subunits

Based on our findings that NSC fate specification is altered by Chd5 deficiency and that Chd5 remodels chromatin both in vitro and in vivo (Li et al, 2014; Quan & Yusufzai, 2014), we hypothesized that Chd5 plays a pivotal role in regulating chromatin and impacting transcription during the early stages of NSC activation. To test this possibility, we visualized DAPI-stained chromatin in wild-type and Chd5-deficient NSCs during differentiation (Fig S2A). In control NSCs, we found that the intensity of the DAPI nuclear signal increased progressively during the initial 48 h of differentiation, indicating that chromatin undergoes dynamic changes during this window (Fig S2B and C). In contrast to controls, Chd5-deficient NSCs had a significantly lower DAPI nuclear signal even before differentiation, suggesting that chromatin was altered (Fig S2B and D). Throughout the entire 48 h of differentiation, the DAPI nuclear signal remained much lower in Chd5-deficient NSCs. Although Chd5-deficient NSCs gradually gained signal intensity during this window, they failed to achieve levels comparable with those of wild-type NSCs, even at 48 h into the differentiation process.

Consistent with this observation, direct visualization of histone clusters by immunofluorescent analyses, combined with structured illumination microscopy, showed that signal intensities of histone H2B spots in Chd5-deficient NSCs were significantly lower both before and 3 h after differentiation (Fig S3A and B). Notably, other physical and distribution properties (i.e., size, distance among clusters, and numbers of neighboring histone clusters) of histone H2B spots in Chd5−/− NSCs were initially unaltered before differentiation but were significantly affected at the 3-h time point, suggesting that Chd5 deficiency exerted its effect on nucleosomes during the differentiation process (Fig S3B and C). These findings suggest that Chd5 deficiency perturbs chromatin during the early stages of differentiation.

Maintenance of chromatin organization is modulated by the polycomb repressive complexes, which methylates lysine 27 (K27) of histone H3 to establish H3K27me3, a key repressive histone modification directly associated with the physical state of chromatin (Williamson et al, 2014; Boettiger et al, 2016). To determine whether the chromatin alterations we observed in Chd5-deficient NSCs were due to modifications of K27 of histone H3, we used immunofluorescent microscopy and western blot analysis (Figs 3A and B, and S3D). These analyses revealed that Chd5−/− NSCs had reduced levels of H3K27me3 and enhanced expression of H3K27ac—a mark associated with transcriptional activation. We also noted that the H3K27 methyl transferase Ezh2 was slightly compromised, which may have contributed to the reduced H3K27me3 levels (Fig 3B). Importantly, RNAi-mediated depletion of the histone demethylase Utx—an enzyme that catalyzes the removal of H3K27me3—rescued the cell fate defects of Chd5−/− NSCs (Fig 3C–E). These observations suggest that aberrant modification of lysine 27 of histone H3 deregulates chromatin dynamics during differentiation, culminating in perturbation of cell fate specification.

To determine if the reduction in H3K27me3 had an effect on global gene expression, we performed RNA-sequencing (RNA-seq) analysis in control and Chd5-deficient NSCs (Fig 3F and Table S1). We found that of the genes that were deregulated in Chd5−/− NSCs (n = 361), the majority (86%) were up-regulated relative to controls, indicating that extensive transcriptional derepression had taken place. Intriguingly, gene ontology (GO) analysis of the deregulated genes revealed a significant enrichment of translation-related and ribosome biogenesis-related GO terms (Fig 3G and Table S2). In fact, of the 84 genes encoding ribosomal proteins, 55% percent were significantly up-regulated in Chd5−/− NSCs (Fig 3F); the remaining genes were either unchanged or were modestly up-regulated (see Table S3). This raised the possibility that Chd5-deficient NSCs had a premature enhancement in translation-associated processes such as ribosome biogenesis and translation initiation. We found

**Figure 1. Chd5 deficiency in NSCs leads to premature activation.**
**(A)** Representative phase contrast images of wild-type (+/+) and Chd5-deficient (Chd5−/−) postnatal day 1 (P1) NSCs grown as adherent cultures (upper images) and neurosphere cultures (lower images). Scale bars = 100 μm (upper images) and 200 μm (lower images). **(B)** Quantification of the size of wild-type and Chd5-deficient NSC neurospheres expressing empty vector (EV) or Chd5 cDNA (Chd5). Data are represented as mean ± SD (n = 3). ns, no significance; **<0.01; ***<0.001; Tukey's multiple comparison test. **(C)** Flow cytometry of EdU-incorporated populations of wild-type and Chd5-deficient NSCs. Data are represented as mean ± SD (n = 4). ***<0.001; unpaired t test. **(D, E)** Flow cytometry of Egfr-positive and Cd133-positive populations in wild-type and Chd5-deficient NSCs. Data are represented as mean ± SD (n = 3). ns, no significance; *<0.05; unpaired t test. **(F, G)** Immunofluorescent images and Western blots of wild-type and Chd5-deficient NSCs, assessed for Chd5 (red), nestin (green), and DAPI nuclear signal (blue) (left panels), or assessed for Pax6, nestin, and β-actin (actin) expression (right panels). Scale bar = 50 μm. **(H, I)** Representative immunofluorescent images of wild-type and Chd5-deficient NSCs, assessed for vimentin (red), nestin (green), and DAPI nuclear signal (blue), and corresponding distribution of mean signal intensities over pixels of nestin and vimentin expression in randomly selected fields (4–5 fields per sample, n = 2–3). Scale bar = 50 μm.

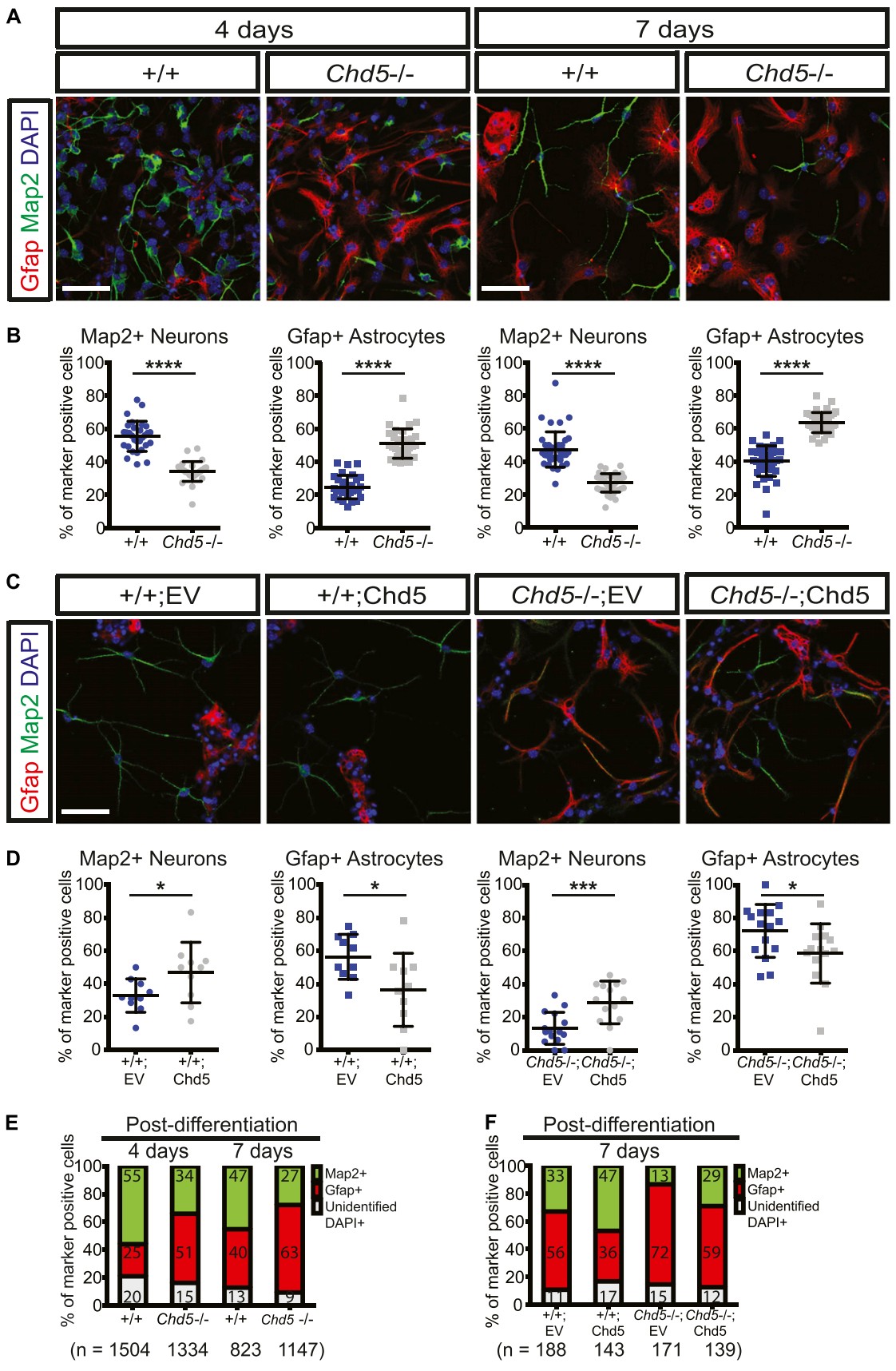

that the large ribosomal RNA precursor (pre-rRNA) in the total RNA pool was significantly down-regulated in Chd5-deficient NSCs, suggesting that premature processing of pre-rRNAs was taking place (Fig S3E, left bars). Despite these changes at the transcriptional level, we did not detect a change in expression of the ribosomal protein Rpl7a via western analysis (Fig S3F). Interestingly, in support of ribosomal biogenesis being augmented in *Chd5−/−* NSCs, quantitative RT–PCR indicated that expression of 18S rRNA—a rate-limiting component required for ribosome assembly—was significantly enhanced (see Fig S3E, right bars). Taken together, these findings indicate that Chd5 affects the chromatin state, deregulation of which leads to extensive transcriptional derepression, including most of the genes encoding ribosomal proteins. This enhanced expression of the translational machinery is correlated with premature activation of NSCs, thereby perturbing neural cell fate.

## Precocious protein synthesis alters dynamic translational control, affecting expression of the proneural factor Mash1

To determine whether the large-scale transcriptional derepression of genes encoding ribosomal proteins we identified in Chd5-compromised NSCs affected overall translational load, we monitored the synthesis of newly translated peptides in control and Chd5-deficient NSCs. We analyzed global protein synthesis by visualizing nascent peptides using the O-propargyl-puromycin (OP-puro)-mediated labeling method (Fig 4A) (Liu et al, 2012). In this assay, cells are briefly pulsed for 30 min with the puromycin analog OP-puro, which becomes incorporated into actively translated peptides; using click chemistry, the newly synthesized peptides are visualized microscopically. Consistent with up-regulation of genes encoding ribosomal proteins, Chd5-deficient NSCs had a significant increase in global protein synthesis relative to controls (Fig 4A and B). This finding indicates that premature activation and transcriptional derepression of ribosomal proteins in Chd5-deficient NSCs lead to enhanced ribosome biogenesis and increased protein synthesis.

We envisioned that enhanced translation might be a normal phenomenon that specifically marks the NSC activation process and, therefore, assessed protein synthesis in wild-type NSCs as they underwent differentiation; we also examined the consequences of increased protein synthesis on the NSC activation process in Chd5-deficient NSCs by tracking temporal changes in OP-puro during differentiation (Figs 4C and D, and S4A and B). Intriguingly, protein synthesis in wild-type NSCs rapidly increased during the first 3 h, declined sharply between 3 and 6 h, and fluctuated in smaller

increments for the next 18 h between 6- and 24-h time points. Notably, the initial surge of protein synthesis peaked just 3 h into the differentiation process (see Figs 4C and D, and S4A). This surge of protein synthesis took place at the beginning of NSC activation and defined the early post-differentiation hours as a critical window. In contrast, the OP-puro signal in Chd5-deficient NSCs was initially higher at the onset of differentiation and then dropped precipitously during the first 3-h window, being reciprocal to the upward trend observed in control cells (see Figs 4C and D, and S4B). Changes in OP-puro signals in both wild-type and Chd5-deficient NSCs were correlated with an increase in phosphorylation of the mRNA cap–interacting eukaryotic translation initiation factor 4E (eIF4E) and a corresponding decrease in the phosphorylated GDP-bound translation inhibitory form of eIF2α, thereby corroborating the observed increase in OP-puro signal was an accurate reflection of an increase in cap-dependent translation initiation (Fig 4E). Notably, the fluctuation of the OP-puro signal observed in Chd5-deficient NSCs between 6- and 24-h post-differentiation were similar to that of wild-type NSCs, suggesting that Chd5 deficiency impinged upon translation during the very first hours of neural differentiation. These observations show the dynamic nature of translation occurring during the NSC activation phase of neural differentiation, and indicate that this process is disrupted in Chd5's absence.

Since the response to differentiation cues occurs so rapidly, we hypothesized that a key cell fate determinant (e.g., a lineage-specific transcription factor) is translated during the earliest stages of differentiation and that disruption of its synthesis alters cell fate potential. To test this possibility, we monitored the nuclear localization and global levels of the proneural transcription factor Mash1—a basic helix-loop-helix transcription factor that maintains homeostasis of neural progenitors and regulates neuronal lineage specification (Ross et al, 2003; Guillemot & Hassan, 2017); we assessed Mash1 during the critical first 3-h window of differentiation (Figs 4F–H and S4C and D). Our analyses revealed that both the frequency of cells with nuclear expression of Mash1 and the overall levels of Mash1, significantly increased during the first 2 h in control NSCs, thus suggesting that the transcript encoding Mash1 (i.e., *Ascl1*) is subject to cap-dependent translation initiation during this critical window of differentiation (Jung et al, 2014). Although the frequency of Mash1+ populations and protein levels in *Chd5−/−* NSCs were initially higher at the onset of differentiation, reflecting premature activation, they rapidly dissipated just 60 min into the differentiation process and remained at lower levels over the course of the assay (Figs 4F–H, S4C and D). The idea that the

**Figure 2.  Chd5-deficient NSCs generate excessive astrocytes at the expense of neurons.**
**(A, B)** Representative immunofluorescent images of wild-type (+/+) and Chd5-deficient (*Chd5−/−*) NSCs (upper panels) that were differentiated for 4 d (left panels) and 7 d (right panels) and analyzed for Gfap (red), Map2 (green), and DAPI nuclear signal (blue), with corresponding quantification (lower panels) of Map2+ neurons and Gfap+ astrocytes in randomly selected fields (10–20 fields per sample). Data are represented as mean ± SD (n = 2–3). ****<0.0001; unpaired *t* test. Scale bar = 50 µm. **(C, D)** Representative immunofluorescent images of wild-type NSCs transduced with empty vector (+/+;EV) or Chd5 cDNA (+/+;Chd5) (left panels) and Chd5-deficient NSCs transduced with empty vector (*Chd5−/−*;EV) or Chd5 cDNA (*Chd5−/−*;Chd5) (right panels), which were differentiated for 7 d and analyzed for Gfap (red), Map2 (green), and DAPI nuclear signal (blue) (upper panels), with corresponding quantification of Map2+ neurons and Gfap+ astrocytes (lower panels) in randomly selected fields (5–8 fields per sample). Data are represented as mean ± SD (n = 2). *<0.05; ***<0.001; unpaired *t* test. Scale bar = 50 µm. **(E)** Distribution of Map2+ neurons and Gfap+ astrocytes of cultures of wild-type and Chd5-deficient NSCs 4 and 7 d post-differentiation; representation of data shown in Fig 2A and B. Numbers indicate mean percentage of each population. n indicates the total number of cells analyzed. **(F)** Distribution of Map2+ neurons and Gfap+ astrocytes of cultures of wild-type and Chd5-deficient NSCs expressing EV or Chd5 cDNA (Chd5) and subject to differentiation for 7 d; representation of data shown in Fig 2C and D. Numbers indicate mean percentage of each population. n indicates the total number of cells analyzed.

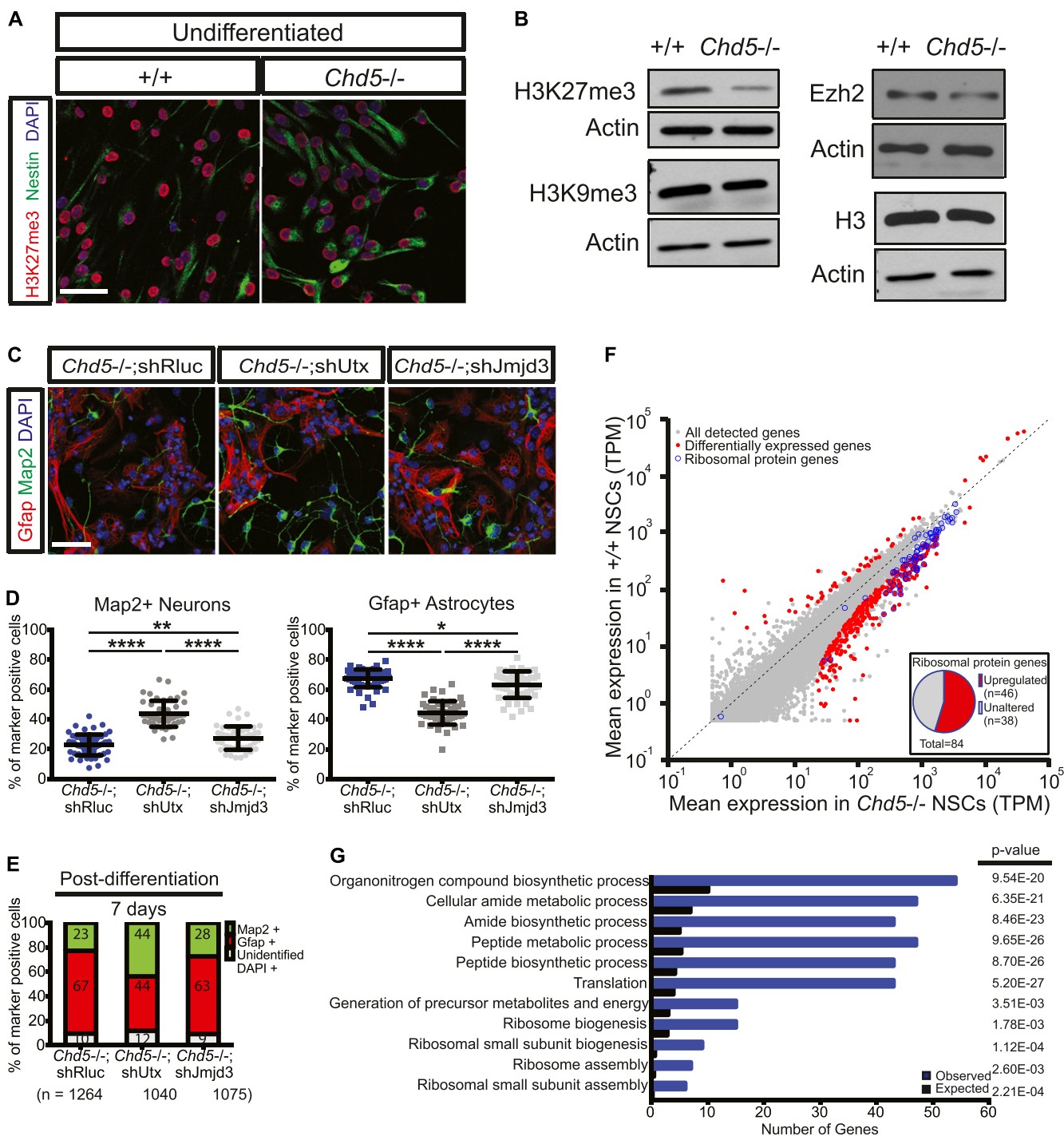

**Figure 3. Chd5 deficiency leads to compromised expression of the repressive histone mark H3K27me3 and up-regulation of ribosomal protein genes.**
**(A, B)** Immunofluorescent images of undifferentiated wild-type (+/+) and Chd5-deficient (*Chd5−/−*) NSCs (left panels) assessed for H3K27me3 (red), nestin (green), and DAPI nuclear signal (blue), and Western blots (right panels) probed for H3K27me3, H3K9me3, Histone H3, Ezh2, and β-actin (actin). Scale bar = 50 μm. **(C, D)** Representative immunofluorescent images of Chd5-deficient NSCs transduced with control shRluc (*Chd5−/−*; shRluc), shUtx (*Chd5−/−*; shUtx), and shJmjd3 (*Chd5−/−*; Jmjd3) that were differentiated for 7 d and analyzed for Gfap (red), Map2 (green), and DAPI nuclear signal (blue) (upper panels), with corresponding quantification (lower panels). Data are represented as mean ± SD (n = 3). *<0.05; **<0.01; ****<0.0001; Tukey's multiple comparison test. Scale bar = 50 μm. **(E)** Distribution of Map2+ neurons and Gfap+ astrocytes in cultures of wild-type and Chd5-deficient NSCs expressing control shRluc, shUtx, and shJmjd3 that were differentiated for 7 d; representation of data shown in Fig 3C. Numbers indicate mean percentage of each population. n indicates the total number of cells analyzed. **(F)** Comparison of steady-state mRNA levels of wild-type and Chd5-deficient NSCs. Mean relative log10-TPM expression values of two replicates for each condition are displayed. Differentially expressed genes are shown in red and ribosomal protein genes are marked with blue circles. Embedded pie chart indicates the fraction of up-regulated ribosomal protein genes (n = 46) among the total number of ribosomal protein genes (n = 84). See also Tables S1 and S2. **(G)** GO terms displaying significant enrichment (>fivefold) of differentially expressed genes in Chd5-deficient NSCs. See also Table S2. TPM, transcripts per million mapped.

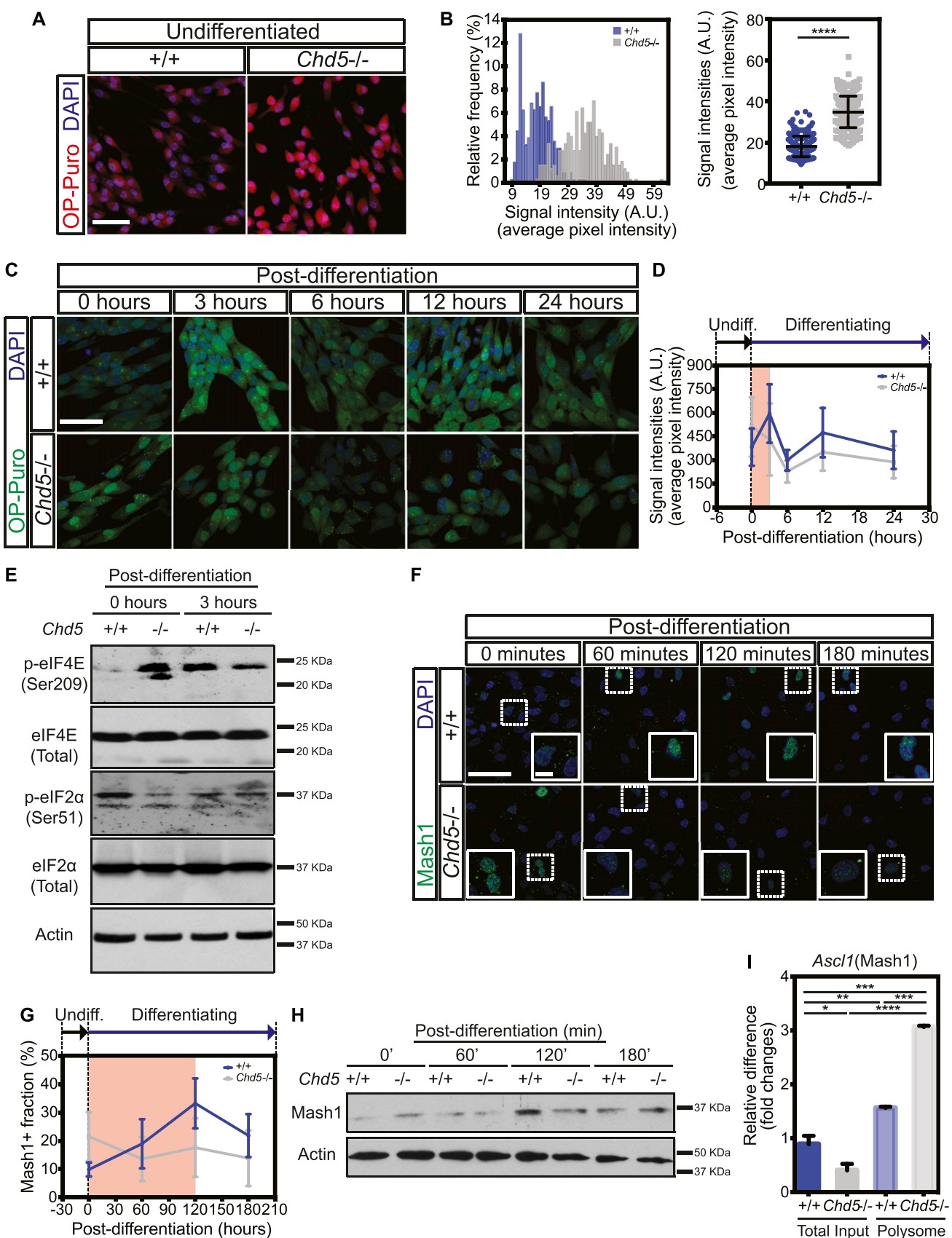

transcript-encoding Mash1 was being prematurely translated in Chd5-deficient NSCs was further supported by our finding that *Ascl1* is depleted in total RNA pools and significantly enriched in actively translating polysome fractions; thus, the transcript-encoding Mash1 is preferentially associated with ribosomes (Fig 4I). These findings illustrate the highly dynamic nature of translational regulation of the key neuronal transcription factor Mash1 during the earliest stages of differentiation, demonstrate that appropriate control of protein synthesis during the onset of differentiation is essential for cell fate specification, and define Chd5 as a critical modulator of these processes.

# Discussion

Transcriptomic profiling of slowly cycling qNSCs and mitotically active aNSCs isolated from the adult mouse brain has identified key intrinsic molecular events that are unique to aNSCs, including a surge of ribosome biogenesis, a shift in the usage of metabolic pathways from glycolysis to oxidative phosphorylation, and up-regulation of a subset of lineage-specific transcription factors (Llorens-Bobadilla et al, 2015; Shin et al, 2015). Although these changes are assumed to collectively underlie the NSC activation process, the temporal coordination and causal relationships between these events in the context of cell fate specification has yet to be clearly understood.

Here, we show that a series of untimely cellular and molecular events triggered by Chd5 deficiency leads to premature activation of NSCs. These perturbations at the very onset of neurogenesis culminate in altered cell fate specification at later stages of neural differentiation, generating an excessive number of astrocytes at the expense of neurons. We discovered a surge of protein synthesis during the initial hours of the neural differentiation process as a crucial event for proper execution of neuronal lineage specification, a shift of which disrupts dynamic translation of the key proneural transcription factor Mash1. These findings provide insight into translational regulation of neuronal cell fate specification and highlight the interplay between transcription and translation as a pivotal mechanism controlling neural cell fate.

## Premature stem cell activation caused by Chd5 loss

Recent identification of aNSCs of the ventricular–subventricular zone of the adult mouse brain reveals the existence of a mitotically active population of radial glial cells with distinct cellular features including increased mitotic activity and enhanced ability to form neurospheres; these cells are known as "activated" NSCs (Codega et al, 2014; Mich et al, 2014). We show that Chd5-deficient NSCs generate larger neurospheres and incorporate EdU at a faster rate, indicating that increased proliferation is a prominent cellular characteristic. Consistent with its previously identified proliferation-suppressive role (Bagchi et al, 2007), Chd5 is sufficient to reverse this phenotype. This enhanced proliferation potential—a defining property of aNSCs—provided a clue that Chd5 deficiency augments the population of activated NSCs.

Activated NSCs express a unique set of cell type–specific markers such as the proliferation-related receptor tyrosine kinase epidermal growth factor receptor (Egfr) and the intermediate filament protein nestin (Codega et al, 2014; Mich et al, 2014). We find that the Egfr+ (i.e., Egfr^High) population is enriched in Chd5-deficient NSCs, further suggesting that expansion of the aNSCs population is driving enhanced proliferation. Consistent with this idea, expression of the aNSC marker nestin is markedly elevated in Chd5-deficient NSCs. These findings indicate that loss of Chd5 leads to a premature shift to the activated stem cell state.

## Premature NSC activation perturbs cell fate specification

Transplantation studies reveal that although both qNSCs and aNSCs are capable of giving rise to neurons, the kinetics of the generation processes differ, suggesting that these two progenitor populations have different cell fate potentials (Codega et al, 2014). Are qNSCs and aNSCs hardwired with distinct lineage specification potentials? Do intrinsic differences in lineage specification potential of progenitors impact neural cell fate? We addressed these questions using Chd5-deficient NSCs—which mirror the cellular properties of aNSCs—and assessed differentiation capacity by quantitating the neurons and astrocytes that were generated. Strikingly, Chd5-deficient NSCs generate more astrocytes at the expense of neurons. These findings indicate that loss of Chd5 in neural progenitors exerts its influence on the intrinsic program of cell fate specification to favor the astrocytic lineage.

Because reintroduction of exogenous Chd5 rescues the cell fate defect, we hypothesize that the untimely NSC activation caused by Chd5 deficiency alters the neural cell fate capacity. We show that the cellular state at the onset of differentiation is functionally tied to the execution of cell fate specification. Neural progenitors in the

**Figure 4. Precocious surge of protein synthesis disrupts dynamic translational regulation of the proneural factor Mash1 during early stages of neural differentiation.**
**(A, B)** Representative immunofluorescent images of wild-type (+/+) and Chd5-deficient (*Chd5*−/−) NSCs (left panels), showing OP-puro–labeled nascent peptides (red) and DAPI nuclear signal (blue), with corresponding distribution and quantification of mean signal intensities over pixels of OP-puro signal (right panels) in randomly selected fields (5 fields per sample). Data are represented as mean ± SD (n = 2). ****<0.0001; unpaired *t* test. Scale bar = 50 $\mu m$. **(C, D)** Representative immunofluorescent images of wild-type and Chd5-deficient undifferentiated (0 h post-differentiation) and differentiated (3, 6, 12, and 24-h post-differentiation) NSCs (left panels), and each image was assessed for OP-puro–labeled nascent peptides (green) and DAPI nuclear signal (blue), with temporal representation of corresponding mean signal intensities over pixels of OP-puro signal at all time points in randomly selected fields (2–6 fields per sample), as plotted in Fig S3A and B. Data are represented as mean ± SD (n = 2). Scale bar = 50 $\mu m$. **(E)** Western blots of undifferentiated wild-type and Chd5-deficient NSCs (0 h post-differentiation) and differentiated cells (3-h post-differentiation), assessed for phosphorylated eIF4E at serine residue 209 (p-eIF4E), unmodified total eIF4E (eIF4E), phosphorylated eIF2$\alpha$ at serine residue 51 (p-eIF2$\alpha$), unmodified total eIF2$\alpha$, and $\beta$-actin (actin) expression. **(F)** Representative immunofluorescent images of wild-type and Chd5-deficient NSCs (left panels) that were undifferentiated (0 min) or differentiated (for 60, 120, and 180 min) and assessed for Mash1 (green) and DAPI nuclear signal (blue). Dashed rectangles indicate the zoomed-in regions, shown in insets. **(G)** Quantification of corresponding frequency of Mash1+ population, displaying distinct nuclear Mash1 expression, in randomly selected fields (5–7 fields per sample) (right panels). Data are represented as mean ± SD (n = 2). Scale bar = 50 $\mu m$ (main image) and 10 $\mu m$ (insets). **(H)** Western blots of wild-type and Chd5-deficient NSCs, assessed for Mash1 and $\beta$-actin (actin) expression at indicated time points. **(I)** Quantification of *Ascl1* (i.e., transcript of Mash1) in total input and polysome fractions of wild-type and Chd5-deficient NSCs. Data are represented as mean ± SD (n = 2). *<0.05; **<0.01; ***<0.005; ****<0.0001; Tukey's multiple comparison test.

developing mouse brain are known to shift from the neuronal phase (i.e., neurogenesis) to the glial phase (i.e., gliogenesis), thus indicating that the outcome of cell fate specification is subject to change, depending on the competence of progenitors (Miller & Gauthier, 2007; Kohwi & Doe, 2013). Thus, cell fate potential (i.e., competence) is dictated by the state of neural progenitors, which is orchestrated by both intrinsic and extrinsic mechanisms that are regulated spatially and temporally.

## Transcriptional derepression and up-regulation of ribosomal protein genes

The chromatin remodeler Chd5 regulates the packaging of the genome during sperm maturation (Li et al, 2014) and is required for migration of differentiating neural progenitors (Egan et al, 2013; Nitarska et al, 2016). At the biochemical level, Chd5 is capable of translocating nucleosomes in vitro, and Chd5 is a component of the nucleosome remodeling and deacetylase complex (Potts et al, 2011; Quan & Yusufzai, 2014). Despite these observations suggesting that the chromatin remodeler Chd5 impacts chromatin structure, the impact of disruption of Chd5 on the transcriptional cascades in NSCs had not been elucidated.

Our observation of premature activation of Chd5-deficient NSCs indicates that intrinsic mechanisms that maintain the ground state of NSCs are perturbed. We reasoned that transcriptional regulation is indispensable for homeostasis of neural progenitors. Indeed, we found transcription to be globally altered by Chd5 deficiency, with the vast majority of differentially expressed genes being up-regulated. Consistent with this broad-scale transcriptional derepression, we noted alterations in DAPI-dense nuclear patterns and H2B patterns in Chd5-deficient NSCs, suggestive of altered chromatin architecture. Thus, the chromatin remodeler Chd5 regulates transcription and, in most of the cases, functions as a transcriptional repressor.

Recent studies of chromatin organization implicate trimethylation of lysine 27 of histone H3 (H3K27me3)—a covalent modification recognized by Cbx components of the polycomb repressive complex 1—in transcriptional repression and chromatin compaction (Zhu et al, 2013; Williamson et al, 2014; Boettiger et al, 2016). Consistent with our findings that transcriptional derepression is a salient feature of Chd5-deficient NSCs, we found enhanced expression of the transcriptionally active histone mark H3K27ac, and concomitant reduced expression of the transcriptional repressive mark H3K27me3. Importantly, perturbation of the H3K27me3-specific demethylase Utx is sufficient to rescue the cell fate specification defect of Chd5-deficient NSCs, indicating that H3K27me3 is functionally important in maintaining appropriate transcriptional control in NSCs and highlighting Chd5's pivotal role in this dynamic process. Taken together, these findings illustrate the crucial role of the Chd5 chromatin remodeler in transcriptional repression in NSCs and maintenance of the progenitor ground state.

## Precocious surge of protein synthesis and compromised translation of the transcript encoding Mash1

The transcriptional derepression of Chd5-deficient NSCs is particularly notable because it causes increased expression of genes encoding ribosomal subunits, suggestive of an increase in ribosome

biogenesis. Intriguingly, enhanced ribosome biogenesis is a key intrinsic molecular feature of aNSCs, in line with our finding that Chd5-deficient NSCs take on an activated state (Llorens-Bobadilla et al, 2015; Shin et al, 2015). We discovered that prematurely activated Chd5-deficient NSCs have a marked increase in nascent peptide synthesis. The question of how this process is temporally regulated during neural differentiation and what functional impact enhanced protein synthesis exerts on translational regulation of key proteins such as lineage-specific transcription factors had not been addressed previously. Using a combination of differentiation assays with imaging-based assessments of nascent protein synthesis, we show that protein synthesis surges just 3 h into the differentiation process in control NSCs, and drops precipitously during the next 3 h, demonstrating the dynamic nature of protein synthesis that normally takes place during the earliest stages of neural differentiation. In contrast, this characteristic surge of protein synthesis does not take place in Chd5-deficient NSCs. Being initially higher than that of controls at the onset, protein synthesis decreases during the first 3 h of the differentiation process; this temporal pattern of Chd5-deficient NSCs is thus reciprocal to that of controls. These findings illustrate that dynamic regulation of protein synthesis during initial stages of differentiation is a crucial event for regulating NSC activation and for maintaining homeostasis of neural progenitor pools.

We reasoned that factors crucial for NSC activation must be preferentially and rapidly translated during this early critical window of cell fate specification. We focused on Mash1, as this transcription factor has a well-documented function in neuronal cell fate specification (Ross et al, 2003), and Mash1 expression in early neural progenitors is required for neuronal differentiation and for the generation of GABAergic-inhibitory neurons in the developing brain (Guillemot & Hassan, 2017). Furthermore, ectopic expression of Mash1 in non-neuronal lineages such as mouse embryonic fibroblasts is sufficient to induce neuronal lineage specification and to generate functional neurons (Chanda et al, 2014). By monitoring changes in Mash1 localization and expression, we found that the proportion of Mash1+ cells and the overall levels of Mash1 in control cells increase significantly just 2 h into the differentiation process, specifically during the window in which we find cap-dependent translation initiation to be enhanced. In contrast, the proportion of Mash1+ cells and the overall levels of Mash1 are initially higher in Chd5-deficient NSCs before differentiation, with Mash1 levels dissipating 2 h into the differentiation process, specifically during the window in which cap-dependent translation tapers down. These findings indicate that perturbation of the dynamics of protein synthesis in Chd5-deficient NSCs takes a toll on translational control and, hence, Mash1 expression.

In summary, we show that the cellular state of progenitor cells dictates the fate of differentiated progenies. Under normal circumstances, NSC activation occurs upon differentiation and lasts for about 12 h. A characteristic surge of protein synthesis and enhanced ribosome biogenesis takes place within hours of NSC activation, heavily influencing translational control and expression of the neuronal lineage regulator Mash1, thereby modulating neuronal cell fate specification. In contrast, absence of the chromatin remodeler Chd5 disrupts the temporal order of events. Chd5-deficient NSCs become activated prematurely, concomitant with transcriptional derepression. This untimely NSC activation perturbs

the very early stages of differentiation, leading to inappropriate translational control that deregulates Mash1 expression, the effect of which collectively manifest as an overproduction of astrocytes at the expense of neurons. Future studies on the mechanisms whereby specific transcripts are normally selected for translation in response to differentiation cues to promote the transition of stem cells from a quiescent to the activated state, and how perturbations in these processes affect tumorigenesis, await to be explored. These findings highlight the immediate and dynamic intrinsic regulatory mechanisms that operate at the post-transcriptional level, which in conjunction with the transcriptional mechanisms regulated by the chromatin remodeler Chd5, determine the ultimate fate of neural progenitors.

## Materials and Methods

### Cell culture

Neural stem/progenitor cells were isolated from mouse brain using the explant culture method (Marshall et al, 2008; Deleyrolle & Reynolds, 2009). *Chd5* (Ref Seq access: NM_029216) was expressed in primary NSCs using retroviral transduction (Keyes et al, 2005; Guo et al, 2009). Knockdown of *Utx* was performed using retroviral constructs expressing shRNAs for *Renilla* luciferase (control) and *Utx*. Morphological features of undifferentiated NSCs, neurospheres, and neurosphere-derived adherent cultures were analyzed by phase contrast microscopy. Neurosphere diameter was assessed using ImageJ software. All animal handling procedures and experimental protocols followed the guidelines of Cold Spring Harbor Institutional Animal Care and Use Committee.

### Immunofluorescent microscopy

Expression of Chd5, nestin, vimentin, Map2, Gfap, Mash1, GFP, H3K27me3, H3K27ac, and actin (i.e., $\beta$-actin) and in NSCs and P1 cortices was assessed using immunofluorescent microscopy (Vestin & Mills, 2013). Protein synthesis was assessed using the OP-puro–based protein assay (Click-iT Plus OPP Protein Synthesis Assay; Molecular Probes). The signal intensities of vimentin, nestin, and OP-puro were quantitated using Volocity software. OP-puro–positive, Mash1-positive, Map2-positive, and Gfap-positive cells were quantitated using the Cell Counter Plugin feature of ImageJ software.

### Flow cytometry

The cellular makeup of NSC populations was examined by assessing fractions of cell surface marker-positive populations (Cd133 and Egfr) using flow cytometry (Codega et al, 2014; Mich et al, 2014). NSC proliferation was measured by EdU (5-ethynyl-2'-deoxyuridine) incorporation (Click-iT Plus Edu; Molecular Probes).

### Molecular analyses

Expression at the protein level was assessed by the Western blot analysis (Gallagher et al, 2001). Global gene expression was assessed by RNA-seq (for detailed materials and methods, see the Materials and Methods section of the Supplementary Information).

### Accession numbers

The accession number for RNA-seq data reported in this manuscript is Gene Expression Omnibus: GSE80583.

## Supplementary Information

## Acknowledgements

We thank Dr Christopher Hammell, Ms Jing Wang, and Dr Gayatri Arun for critical insights and sharing experimental protocols. We thank the members of the Mills laboratory for helpful suggestions and insights. This work made use of the Cold Spring Harbor Laboratory Shared Resources, which are funded in part by the Cancer Center Support Grant 5P30CA045508. This project was supported by the Office of the Director, National Institutes of Health, under award numbers R01CA127383 (to AA Mills), R01CA190997 (to AA Mills), and R21OD018332 (to AA Mills). Support also came from the Stanley Foundation (to AA Mills).

### Author Contributions

D-W Hwang: conceptualization, data curation, formal analysis, and writing—original draft, review, and editing.
A Jaganathan: conceptualization, data curation, formal analysis, and writing—review and editing.
P Shrestha: conceptualization and data curation.
Y Jin: data curation, software, formal analysis, and writing—review and editing.
N El-Amine: data curation.
SH Wang: conceptualization.
M Hammell: data curation, software, and writing—review and editing.
AA Mills: conceptualization, resources, data curation, formal analysis, supervision, funding acquisition, and writing—original draft, review, and editing.

### Conflict of Interest Statement

The authors declare that they have no conflict of interest.

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
