## [Reviewer comments · Life Science Alliance]

Chromatin-Mediated Translational Control is Essential for Neural Cell Fate Specification

Dong-Woo Hwang, Anbalagan Jaganathan, Padmina Shrestha, Ying Jin, Nour El- Amine, Sidney H. Wang, Molly Hammell, and Alea A. Mills
DOI: 10.26508/lsa.201800016

Review timeline:

Submission Date:	19 December 2017
Editorial Decision:	13 February 2018
Revision Received:	4 August 2018
Editorial Decision:	8 August 2018
Accepted:	14 August 2018

Report:

(Note: Letters and reports are not edited. The original formatting of letters and referee reports may not be reflected in this compilation.)

1st Editorial Decision

13 February 2018

Thank you for submitting your manuscript entitled "Chromatin-Mediated Translational Control is Essential for Neural Cell Fate Specification" to Life Science Alliance. Please excuse again the delay in getting back to you, I had to give the reviewers more time and the peer-review took longer than usual. The manuscript was assessed by two expert reviewers, whose comments are appended to this letter. We invite you to submit a revision if you can address the reviewers' key concerns, as outlined here.

As you will see, the reports are very consistent, but while the individual points raised can be in principle addressed in a revision, quite a bit of work is needed to address key technical concerns. I would therefore be keen to discuss the revision with you further, and I invite you to get back to me with a preliminary response to the concerns raised.

Importantly, we would expect for acceptance a satisfactory revision addressing all comments by referee #2. mTOR inhibition to check for similar effects and Mash1 protein stability assessment are not mandatory for publication. Point 5 is also a concern that is raised by referee #3, and we therefore strongly encourage you to perform polysome profiling for mash1. Regarding referee #3's other concerns, we would expect the issue with DAPI staining as readout for chromatin state to be resolved, and H3K27me3 should be monitored. You should also comment and change your text/alter the analysis of existing data to address this referee's other concerns.

REFEREE REPORTS

Reviewer #2 (Comments to the Authors (Required)):

In this article, Hwang et al provide evidence suggesting a role for Chd5 in transcriptional regulation of genes encoding ribosomal proteins and subsequent reprogramming of mRNA translation in neural cell fate specification. Specifically, the authors show that the loss of Chd5 favors astroglial differentiation while disfavoring neuronal differentiation. This appears to be correlated with the alterations in Mash1. Overall, this study provides insights into mechanisms which coordinate different steps of regulation of gene expression (chromatin remodeling, transcription and translation) which underpin differentiation of neural stem cells. I therefore find the authors' findings significant and of broad interest. However, I thought that there are several issues that warrant rectification. My specific comments are listed below:

Major comments:

1. The mechanism by which changes in transcription of genes encoding ribosomal proteins selectively impact on mRNA translation remains unclear. Of note, most of the mRNAs encoding ribosomal proteins contain 5'TOP motif and are thus regulated at the level of translation. Hence, notwithstanding relatively long half-lives of ribosomal proteins, the authors should verify whether the changes in steady-state mRNA levels correspond to changes in the protein at least for a few of the ribosomal proteins that appear to be regulated in a Chd5-dependent manner.

2. Is the induction of transcription of the genes encoding ribosomal proteins coordinated with increased rDNA transcription?

3. In addition to Mash1, the authors should also include more proteins which are expected to be translationally affected, as well as those that are non-affected by the Cdh5 status in the cell. This is important as it may help establish the selectivity of translational changes.

4. The quality of p-4E-BP1 blot in figure 4G is poor, and for me it was hard to appreciate any significant differences in 4E-BP1 phosphorylation, in particular in the absence of the band shifts in total 4E-BP1 blots. It is also thought that 4E-BP2 is a major 4E-BP family member in the brain, so perhaps it is also worth probing for 4E-BP2. Notably, p-Ab recognizing residues T37/46 will recognize both 4E-BP1 and 2. Are the other mTORC1 targets (e.g. S6 kinases) also regulated in a similar manner as 4E-BPs? Can observed effects be phenocopied using mTOR inhibitors?

5. The differences in translation of Mash1 mRNA should be directly assessed by either labeling/IP or preferably polysome profiling as this is one of the major conclusions of the manuscript. Polysome profiling will also help estimate the engagement of the ribosomes in translation in relation to Cdh5 status. Finally, total mash1 mRNA levels as well as protein stability should be determined as a function of Cdh5 status in the cell.

Minor comments:

1. What is the mechanism of Ezh2 reduction in Chd5 cells (figure 3B)? Is the reduction in Ezh2 protein levels in Chd5^{-/-} cells due to the transcriptional or post-transcriptional mechanisms (perhaps translation)?

2. Phosphoacceptor sites recognized by Abs should be included in figure panels.

3. The authors should discuss how Chd5 exerts selective effects on translation (e.g. specialized ribosomes etc).

I hope that the authors will find my comments and suggestion constructive and of sufficient paths.

Sincerely

I/Topisirovic

Reviewer #3 (Comments to the Authors (Required)):

In this work, Hwang et al. described the role of CHD5 in neural cell fate specification. They provided data showing that Chd5 deficiency enhances neural stem/progenitor cells (NSCs) proliferation and Chd5-deficient NSCs undergo altered cell fate decisions, favoring the astroglial over the neuronal lineage. The authors suggested that these defects are due to chromatin decompaction, downregulation of EZH2 and H3K27me3 levels, enhancement of ribosome biogenesis and translation.

This work has some potential interesting findings. The altered cell fate decision Chd5-deficient NSCs described in the first two figures is convincing and well done. However, all attempts to provide a molecular explanation of this phenotype are weak and often overinterpreted. Moreover, I

found that a Discussion of 10 pages (!) is quite long... many of the discussed points are redundant since they were already present in the result section.

Major points.

The interpretation of DAPI staining to measure chromatin condensation is not correct. DAPI intercalates with AT rich sequences and the background staining between cells (excluded the foci) should be the same and this was not the case. To imply alteration of chromatin compaction in *Chd5*-deficient NSCs the authors should have measured the amounts and size of DAPI stained foci, which represent centric and pericentric heterochromatin (in mouse, major and minor satellites) and not the overall DAPI staining. From the pictures provided in EV2, I do not see any remarkable changes in heterochromatin compaction or organization.

The authors provided data that implicated H3K27me3 in the defects of cell fate decision of *Chd5*^{-/-} NSCs. However, the authors have to be careful in linking their DAPI analysis with H3K27me3. The authors stated that "Maintenance of chromatin compaction is modulated by the Polycomb Repressive Complexes (PRC) and the repressive histone modification H3K27me3". This is true. However, the role of PcG and H3K27me3 in chromatin compaction cannot be monitored by DAPI staining (DAPI foci which represent constitutive heterochromatin depend on H3K9me3 and not on H3K27me3) but only with sophisticated technology such as HiC or super-resolution imaging. The data clearly showed that H3K27me3 and EZH2 levels decrease in *Chd5*-deficient NSCs.

Interestingly, H3K9me3 levels are also shown in Fig. 3B but not described. The signal of this image is close to saturation but it seems to me that H3K9me3 signal decreases in *Chd5*^{-/-}. Are there some changes? The authors rescued the cell fate defects of *Chd5*^{-/-} NSCs at 7 days post-differentiation (high expression of *Map2*) by knockdown of *Utx*, which demethylates H3K27me3. However, downregulation of *Jmjd3*, which can also demethylate H3K27me3, had no effect. These results are in Fig. 3D but not discussed in the text. Is *Jmjd3* expressed in NSCs?

The RNAseq analysis showed global changes in gene expression. How many genes are deregulated? Which fold changes and P values has been chosen?

The GO analysis indicated upregulation of many ribosomal genes. The authors interpreted these results as enhanced ribosome biogenesis " p. 10: This enhancement of ribosome biogenesis correlates with premature activation of NSCs, thereby perturbing neural cell fate." This is an overinterpretation of the data and no experimental evidence is provided. Ribosome biogenesis is a complex pathway and an increased level of ribosomal proteins does not necessarily imply enhanced ribosome biogenesis. The authors should perform a ribosomal profile and validate by western the upregulation of some of the ribosomal proteins.

The authors analyze translation efficiency through labeling with O-propargyl-puromycin (OP-Puro) and determine that *Chd5*^{-/-} NSCs have a significant increase in global protein synthesis relative to wildtype controls. Less clear is how during differentiation global protein synthesis decreases in *Chd5*^{-/-} respect to wild type as well as the levels of *Mash1*. Is the control of *Mash1* expression at protein or at transcription levels? Some more experiments should have been dedicated to define the distinction between transcription and translation control. It is also quite unclear the link between *Mash1* and cap-dependent translation initiation. The only data is a western where I do not see any increase of phosphorylation at 4EBP1. Moreover, even in the case of elevated phosphorylation at 4EBP1 this should affect many other proteins. Therefore I do not see how these results can demonstrate the dynamic of translational regulation as an important determinant during differentiation without having analyzed whether a factor such as *Mash1* is transcriptionally regulated.

Overall, I think that this work would have benefit to analyze in more details one single pathway instead of presenting few and insufficient data on four different complex processes (chromatin organization, Polycomb regulation, ribosome biogenesis and regulation of translation).

1. Preferential translation of the transcript encoding *Mash1* in *Chd5*-deficient NSCs

Both Reviewers raised the interesting question of whether the *Ascl1* transcript (which encodes the *Mash1* protein) is preferentially translated in *Chd5* deficient NSCs.

To address this possibility, we compared the relative expression of *Ascl1* in the polysome fraction (where actively translating transcripts are enriched) of *Chd5*-deficient NSCs with that of wildtype control, using qRT-PCR (Fig 4I).

Our analyses indeed indicate that *Ascl1* transcript is clearly more enriched in the polysome fraction of *Chd5* compromised NSCs, suggesting that translation of *Ascl1* is augmented. This is consistent with the increased levels of *Mash1* in *Chd5*-deficient NSCs (Fig 4H). Importantly, this enhancement in translation of *Ascl1* and the subsequent increase in *Mash1* protein are consistent with: our findings that: 1) our initial finding that global translation is increased, 2) our new finding that ribosome biogenesis is enhanced, and 3) our new finding that there is a marked increase in the phosphorylated 'translation activating' form of eIF4E, as well as a decrease in the phosphorylated GDP-bound 'translation inhibitory' form of eIF2 α in *Chd5*-deficient NSCs (Fig S3E & Fig 4E, please see points #2 and #4 for more details on alterations in expression of translation- modulating proteins and rRNA's, respectively).

Strikingly, although our initial RNA-sequencing analysis with a stringent cutoff did not detect *Ascl1* as being differentially expressed in *Chd5*^{-/-} NSCs (Table S1), qRT-PCR in the total RNA pool indicate that *Ascl1* transcript levels are lower in cells lacking *Chd5*, suggesting that the robust enhancement of translation overrides transcriptional downregulation (Fig 4I).

2. Engagement of translation in *Chd5*-deficient NSCs

A major question raised in this review was whether proteins modulating translation are really altered in *Chd5*-deficient NSCs. We had initially attempted to answer this question by performing westerns for phosphorylation of 4EBP1, a modification which functions as a tight switch that prevents 4EBP1 from binding and inhibiting the eukaryotic translation initiation factor 4E (eIF4E), leading to activation of cap-dependent translation initiation. We completely agree with reviewer #2 that the western blots shown in the original submission were not very convincing.

To strengthen our claim that proteins affecting translation are altered in *Chd5* deficient cells, we decided to monitor direct readouts for translation initiation: the active translation form of cap-interacting protein eIF4E (i.e. eIF4E that is phosphorylated at Ser209) and the translation inhibitory form of methionyl initiator tRNA-interacting eIF2 α (i.e. eIF2 α that is phosphorylated at Ser51).

Strikingly, western analyses reveal that phosphorylated eIF4E is markedly enhanced, whereas phosphorylated eIF2 α is significantly reduced in undifferentiated *Chd5*-deficient NSCs (Fig 4E). Consistent with our findings that global protein synthesis drops upon differentiation (Fig 4C-D), the differences in phosphorylated eIF4E and phosphorylated eIF2 α were reversed and mitigated, respectively, 3 hours into the differentiation process (Fig 4E). These findings further support our initially proposed model that *Chd5* deficiency causes premature translation.

In addition to factors that modulate translation being altered, we also found that 18S rRNA is enhanced in *Chd5*^{-/-} NSCs, further indicating that there is enhanced ribosomal biogenesis (Fig S3E, please see point #4 for more details).

We appreciate the valuable suggestions of the reviewers, as it now allows us to convincingly establish that ribosome biogenesis is increased, that phosphorylated eIF4E that induces cap-dependent translation initiation is enhanced, and that phosphorylated eIF2 α that inhibits the turnover of the translation initiation machinery is compromised, in the context of *Chd5* deficiency.

3. Westerns to show additional proteins that are enhanced/not enhanced

Reviewer #2 would like to know whether there are other proteins besides *Mash1* that are affected in *Chd5*^{-/-} NSCs, or whether global regulation of translation is occurring.

To determine whether certain proteins are regulated at the translational level in cells lacking *Chd5* while others are not, we examined a panel of NSC-specific markers in control and *Chd5*-deficient NSCs (Fig 1G). This examination revealed that while the general NSC marker *Pax6* is decreased, the activated NSC marker *Nestin* is enhanced, indicating that protein expression is dictated by the cellular state, and further supporting our initial observation that *Chd5* deficiency leads to premature activation in NSCs (Fig 1).

In addition, we also probed for markers for intermediate progenitors (*Tbr2*) and the neuronal precursors (*Tbr1*) under both proliferating and differentiated conditions (Fig S1G). These analyses indicate that *Tbr2* is not affected under either condition, whereas *Tbr1* is compromised upon differentiation.

These observations clearly show that there are specific proteins whose expression is selectively affected by the absence of Chd5, whereas others are unaffected. We edited the Discussion to point out that future studies could be performed to uncover the molecular mechanisms whereby specific transcripts are selected for translation upon neural stem cell activation (see final paragraph).

4. qRT-PCR to assess transcription of rRNA

A great question posed by one of the referees was whether the augmented translation of Chd5-deficient NSCs correlates with enhanced expression of RNA components of the ribosome. To determine whether expression of ribosomal RNAs are altered in Chd5 deficient NSCs, we used qRT-PCR to assess expression of the large ribosomal RNA precursor (pre-rRNA). These analyses indicated that expression of large pre-rRNAs are decreased in total RNA sample of Chd5-deficient NSCs compared to that of the wildtype control (Fig S3E). While this was somewhat unexpected, we hypothesize that this is due to the fact that this assay for rRNAs measures pre-ribosomal RNAs that have not yet been processed; therefore, the observed decrease in these pre-rRNAs could reflect premature processing of rRNAs in Chd5-compromised NSCs. To test this idea, we assessed expression of the 18S processed product of the large pre-rRNA. Indeed, as this Reviewer had asked us to test, there is an increase in processed rRNA in Chd5 deficient NSCs, in agreement with ribosome biogenesis being enhanced (Fig S3E).

5. The extent that ribosomal proteins are regulated at the level of transcription and/or translation

The reviewers asked whether ribosomal subunits are being regulated primarily at the transcriptional level, or at the translational level as well.

To address this question, we assessed expression of the protein component of the small 40S ribosomal subunit (i.e. Rps6) and that of the large 80S ribosomal subunit (i.e. Rpl7a) in control and Chd5-deficient NSCs under both proliferating and differentiated conditions (Fig S3F).

Our western blot analyses indicate that there is no observable alteration in expression of these ribosomal proteins in Chd5-deficient cells. A possible explanation is that ribosomal proteins are so abundant that their changes in expression are difficult to detect by western blot analysis. Whereas we did not find evidence for translation regulation of ribosomal protein expression, we did detect an increase in 18S rRNA transcript, suggesting that this rate limiting component essential for ribosome assembly (and/or the initiation factors that serve as ‘switches’ for translation; pls see point #2, above) contribute to the increase in protein synthesis in the context of Chd5 deficiency.

6. The effect of Chd5 loss on chromatin

Questions regarding the degree to which chromatin is affected by Chd5 deficiency were brought up by the expert reviewers. Specifically, several suggestions to more clearly demonstrate that chromatin is less compact in Chd5-deficient NSCs were made, such as performing immunofluorescent analyses for H3K27me3 and H3K27ac.

To make a more convincing case that Chd5 deficient NSCs have altered chromatin, we took an imaging-based approach to visualize eGFP-labeled endogenous histone variant H2B using high resolution structured illumination microscopy (SIM). We assessed eGFP-H2B using immunofluorescence and an antibody specific for eGFP, acquired images, used binary transformation, and analyzed distribution patterns of H2B signal spots, which reflect the distribution of nucleosomes (Fig S3A-C).

These extensive analyses revealed that the histone H2B spots in Chd5-deficient NSCs were significantly lower both at the onset of differentiation and 3 hours into the differentiation process, thereby supporting our initial observation of reduction in DAPI signal intensities (Fig S2 and Fig S3B).

In addition, we used immunofluorescence to monitor H3K27me3 as suggested by the Reviewers, revealing that that expression of this transcriptional repressive mark is compromised in Chd5-deficient NSCs (which we also show using western analyses) (Fig 3A and Fig 3B). Furthermore, we assessed expression of the transcriptional activating mark H3K27ac, as methylation and acetylation both occur on the same residue, but with opposing roles on chromatin and transcription. These analyses indicate that lysine 27 of histone H3 has compromised methylation and enhanced acetylation, in agreement with the transcriptional de-repression we observed.

In addition to these additional data regarding the alteration in histone marks, we also edited the text to focus on these features, rather than on chromatin compaction.

Significant changes to the text were made to clarify the points made by the reviews, to shorten the discussion as suggested, and to incorporate the extensive amount of new data that we added.

2nd Editorial Decision

8 August 2018

Thank you for submitting your revised manuscript entitled "Chromatin-Mediated Translational Control is Essential for Neural Cell Fate Specification". As you will see below, reviewer #2 saw your revised manuscript again and now supports publication of your work in Life Science Alliance. We would thus be happy to publish your paper in Life Science Alliance pending final revisions necessary to meet our formatting guidelines.

REFeree REPORTS

Reviewer #2 (Comments to the Authors (Required)):

I thought that the authors have addressed my concerns and comments in a satisfactory manner.

I/Topisirovic